# Development of a Novel Evaluation Method for Endoscopic Ultrasound-Guided Fine-Needle Biopsy in Pancreatic Diseases Using Artificial Intelligence

**DOI:** 10.3390/diagnostics12020434

**Published:** 2022-02-08

**Authors:** Takuya Ishikawa, Masato Hayakawa, Hirotaka Suzuki, Eizaburo Ohno, Yasuyuki Mizutani, Tadashi Iida, Mitsuhiro Fujishiro, Hiroki Kawashima, Kazuhiro Hotta

**Affiliations:** 1Department of Gastroenterology and Hepatology, Graduate School of Medicine, Nagoya University, 65 Tsurumai-cho, Showa-ku, Nagoya 4668550, Japan; hsuzuki@med.nagoya-u.ac.jp (H.S.); eono@med.nagoya-u.ac.jp (E.O.); y-mizu@med.nagoya-u.ac.jp (Y.M.); iidatyuw@med.nagoya-u.ac.jp (T.I.); 2Department of Electrical and Electronic Engineering, Faculty of Science and Technology, Meijo University, Nagoya 4688502, Japan; 203427018@ccmailg.meijo-u.ac.jp (M.H.); kazuhotta@meijo-u.ac.jp (K.H.); 3Department of Gastroenterology, Toyohashi Municipal Hospital, Toyohashi 4418570, Japan; 4Department of Gastroenterology and Hepatology, Graduate School of Medicine, The University of Tokyo, Tokyo 1138655, Japan; mtfujish@gmail.com; 5Department of Endoscopy, Nagoya University Hospital, Nagoya 4668550, Japan; h-kawa@med.nagoya-u.ac.jp

**Keywords:** artificial intelligence, endoscopic ultrasound-guided fine-needle biopsy, pancreatic diseases, deep learning, contrastive learning, stereomicroscope

## Abstract

We aimed to develop a new artificial intelligence (AI)-based method for evaluating endoscopic ultrasound-guided fine-needle biopsy (EUS-FNB) specimens in pancreatic diseases using deep learning and contrastive learning. We analysed a total of 173 specimens from 96 patients who underwent EUS-FNB with a 22 G Franseen needle for pancreatic diseases. In the initial study, the deep learning method based on stereomicroscopic images of 98 EUS-FNB specimens from 63 patients showed an accuracy of 71.8% for predicting the histological diagnosis, which was lower than that of macroscopic on-site evaluation (MOSE) performed by EUS experts (81.6%). Then, we used image analysis software to mark the core tissues in the photomicrographs of EUS-FNB specimens after haematoxylin and eosin staining and verified whether the diagnostic performance could be improved by applying contrastive learning for the features of the stereomicroscopic images and stained images. The sensitivity, specificity, and accuracy of MOSE were 88.97%, 53.5%, and 83.24%, respectively, while those of the AI-based diagnostic method using contrastive learning were 90.34%, 53.5%, and 84.39%, respectively. The AI-based evaluation method using contrastive learning was comparable to MOSE performed by EUS experts and can be a novel objective evaluation method for EUS-FNB.

## 1. Introduction

Endoscopic ultrasound-guided fine-needle aspiration (EUS-FNA) has been widely used as a technique to collect pancreatic tissue [1]. With the development of treatment options, including neoadjuvant chemotherapy, immune checkpoint inhibitors, and gene panel tests, the importance of tissue collection during pancreatic cancer treatment has been increasing. Recently, several new needles with novel needle tip shapes have been developed, making it possible to collect a larger amount of tissue [2,3]. These new core needles are used in a technique called endoscopic ultrasound-guided fine-needle biopsy (EUS-FNB). Various attempts have been made to evaluate whether proper specimens are being collected under EUS guidance. In 2011, the usefulness of rapid on-site cytology (ROSE) [4] was reported as a specimen evaluation method during EUS-FNA. Recently, a touch imprint cytology technique was reported for EUS-FNB specimens, which allows to obtain both cytological and histological specimens at the same time with the same needle, as well as to perform ROSE [5]. This technique provided comparable samples to those of EUS-FNA-standard cytology and combined the benefits of cytology and histology for the evaluation. However, the number of facilities that can perform ROSE is limited, and this approach does not necessarily lead to a reduction in procedure time. In addition, the advent of core needles has allowed for an increase in the amount of tissue that can be obtained with a smaller number of needle passes, and some reports suggest that ROSE may not be needed to reduce the number of needle passes in the era of EUS-FNB [6,7,8]. Moreover, the usefulness of macroscopic on-site evaluation (MOSE) has been reported as an alternative to ROSE [9]. Chong et al. [10] reported that MOSE provided the same diagnostic performance as EUS-FNA in the absence of ROSE and reduced the number of punctures. In general, MOSE results are considered positive when whitish tissue (core tissue) can be seen macroscopically in the obtained specimen. However, the judgement is often made by the endosonographer and is largely subjective based on experience level.

Recently, there has been a remarkable development of artificial intelligence (AI) using convolutional neural networks (CNNs) [11]—a deep learning method—in the field of image recognition in medicine [12,13]. This technology has also been applied to the field of gastroenterology and endoscopic procedures, including for the diagnosis of oesophageal cancer [14], gastric cancer [15], colorectal polyps [16], and the EUS-based diagnosis of pancreatic disease [17].

A previous study reported the diagnosis of pancreatic ductal adenocarcinoma (PDAC) using a deep learning model based on pathological specimens obtained by EUS-FNB [18]. However, to the best of our knowledge, there are no studies that have examined the usefulness of AI in predicting the diagnosable material for histology using fresh specimens. In this study, we aimed to develop a new AI-based method that can be an alternative to MOSE for evaluating EUS-FNB specimens in pancreatic diseases using deep learning and contrastive learning.

## 2. Materials and Methods

### 2.1. Study Design

This was a retrospective study conducted as a medical-industrial collaborative project between the Department of Gastroenterology and Hepatology at Nagoya University Hospital and the Faculty of Science and Technology at Meijo University. It was performed with the approval of the ethics committee of each institution. The content of the research was described, and contact information was provided in an opt-out format on the website of our hospital for patients who did not wish to participate (approval number: 2019-0310). The study was performed in accordance with the ethical standards stated in the 1964 Declaration of Helsinki and its later amendments or comparable ethical standards associated with Grant-in-Aid for Scientific Research (grant no. JP20K12689) support.

### 2.2. Patients

This study consists of two stages: the first stage uses deep learning, and the second stage uses contrastive learning. For the deep learning stage, we reviewed 98 specimens from 63 patients in whom EUS-FNB was performed for pancreatic diseases using a 22 G Franseen needle (Acquire, Boston Scientific Co., Natick, MA, USA) between September 2019 and October 2020 in Nagoya University Hospital, and all specimens obtained were photographed by a stereomicroscope (SZX12, Olympus Co., Ltd., Tokyo, Japan) immediately after specimen collection. There is no clear rationale for the sample size setting because there are no previous studies that have examined the same issue. Therefore, we first decided to use the specimens obtained by EUS-FNB performed at our hospital during a one-year period. Then we added cases up until January 2021, because the performance is expected to increase with the number in AI analysis, and a total of 173 specimens from 96 patients were reviewed for the later study of deep learning and contrastive learning stage.

### 2.3. EUS-FNB Procedure

The EUS procedure was performed by two experts with more than 10 years (EO and TIs) of experience or by trainees supervised by the experts using a linear-array endoscope (GF-UCT260, Olympus Co., Ltd., Tokyo, Japan) and an ultrasound system (EU-ME2, Olympus Co., Ltd., Tokyo, Japan). While the patient was under conscious sedation, the EUS scope was inserted orally. The lesion was carefully observed in B-mode first and then in colour Doppler mode before the puncture was performed to confirm that no major vessels were in the needle pathway. After the needle was inserted into the lesion, the stylet was slowly withdrawn (dry slow-pull technique) as the sample was obtained; this was repeated for all needle passes. The number of passes was determined based on whether ROSE confirmed the presence of tumour cells, with a maximum of three passes.

### 2.4. Specimen Processing for EUS-FNB

The specimen was extruded from the needle onto a Petri dish using saline. The liquid components around the specimen were then aspirated with a syringe and prepared for ROSE and cytology. The aspirated liquid component was diluted to 6 mL with saline. Smears were prepared by processing with Autosmear (Sakura Finetech Japan, Co., Ltd., Tokyo, Japan) at 1500 rpm for 5 s, promptly spray fixed (Melcofix^®^, Merck KGaA, Darmstadt, Germany), and stained with the ultrafast Papanicolaou (UFP) method. Microscopic evaluation was performed inside the endoscopy suite by an experienced cytologist, and the presence or absence of cell components was immediately reported to the endosonographer. The remaining solid specimens were immediately observed under a stereomicroscope, and images were photographed. The specimens were then placed in formalin solution for histological examination. All specimens were processed per needle pass, and slides with haematoxylin and eosin (HE) staining were also made per needle pass.

### 2.5. MOSE and Imaging EUS-FNB Specimens

A high-end zoom stereomicroscope, SZX12, was used for stereomicroscopic observation. The magnification range was from 7× to 90× (zoom ratio 12.86) with an aperture mechanism that allowed for a deeper depth of field. To evaluate the specimens under the same conditions, the observation screen was set up so that the vertical width was 2 cm with a scale of 1-mm increments underneath (Figure 1). The specimen was then observed with nothing in the background, and a single expert endosonographer (TIs) who has performed more than 300 EUS-FNB procedures, performed MOSE. A specimen was defined to have positive MOSE results if it contained a portion recognisable as whitish core tissue, with or without the presence of reddish blood clots. A specimen was defined to have negative MOSE results if it contained little or no core tissue or only reddish blood clots. The images used for MOSE were captured as a JPG file by the digital camera attached to the stereomicroscope and were sent for AI-based analysis. To assess the reliability of MOSE, the stereomicroscopic images were independently reviewed after the EUS-FNB procedures by another expert endosonographer (HS) who had performed more than 200 EUS-FNB procedures and was blinded to the patient history and clinical, radiologic, and histologic information.

### 2.6. Histology Evaluation

After being fixed in formalin, the specimens were embedded in paraffin, sectioned, and subjected to HE staining and appropriate immunostaining according to the suspected diagnosis. Histological diagnosis was performed by 2 pathologists who specialise in the pancreatobiliary field at Nagoya University Hospital. The final diagnosis was based on the surgical specimen or the clinical course consistent with the results of EUS-FNB with a minimum 6-month follow-up. The histological diagnoses were classified into one of the following 5 categories: malignant, suspicious, atypical, benign, and inadequate. For tumorous lesions such as pancreatic cancer, it was considered to be correctly diagnosed if they were malignant or suspicious. For benign lesions such as mass-forming pancreatitis, in addition to showing atypical or benign characteristics on histology, a diagnosis was made if the lesion was confirmed to not worsen during the 6-month follow-up period. To evaluate the amount of core tissues obtained for each specimen, the total area of the core tissue was marked and measured under a photomicroscope using imaging software (CellSense, Olympus Co., Ltd., Tokyo, Japan) based on our previous reports [2,19,20]. The microscope was connected to a computer with CellSense software installed, and a digitised image of the HE-stained slide optimised for quantitative analysis could be directly captured using the software. The specimen was imaged at low magnification so that the whole specimen was included in a single image, and the area of the intact core tissues was measured manually. The investigator who performed the measurements was blinded to any clinical characteristics or histological results (Figure 2).

### 2.7. AI-Based Evaluation Using Deep Learning

For deep learning, we used AlexNet [21], an image recognition neural network. AlexNet consists of 3 convolutional layers, 2 pooling layers, and 3 fully connected layers. The size of the input image was 224 × 224 pixels, and the output was a one-dimensional vector with 2 elements. In AlexNet, a square image is divided into upper and lower branches and then combined in the final all-combining layer. Of the 98 specimens included in this study, 81 were classified as histologically diagnosable and 17 as histological nondiagnosable, and we examined whether AI methods using deep learning could predict the diagnosable material for histology based on stereomicroscopic images (Figure 3). To address the small amount of imaging data, augmentation was performed by flipping and rotating the existing imaging data clockwise. The 81 histologically diagnosable images were augmented to 162 images by 90-degree rotation, the 17 nondiagnosable images were augmented to 136 images by flipping plus 3-way rotation, and a total of 298 images were used for analysis (Figure 4) [22]. For the deep learning analysis, we divided the dataset into 3 sets to ensure that the small number of histologically nondiagnosable images were all evaluated. Training and validation were performed for each data set, and the final accuracy was calculated by averaging the accuracy based on the 3 data sets.

### 2.8. AI-Based Evaluation Using Contrastive Learning

We used an unsupervised representation learning method, such as SimCLR [23] for contrastive learning. Unsupervised representation learning is a concept that requires drastically less training by humans and allows the AI to learn more on its own. Contrastive learning allows the features of the same image with different processing to approach each other and the features of different images to repel each other. We trained several networks simultaneously. In one such network, the AI learned the correspondence between the HE-stained image and the image with the marked core tissue and detected which pixels corresponded to the tissue. As a result, we achieved an overlap rate (intersection over union, IoU) of 89.6% between the HE-stained images and the marked images. Moreover, the tissue area was predicted based on the segmentation of the core tissue. The contrastive learning model was also trained to approach the features of the stereomicroscopic images and HE-stained images, and we investigated whether linking the two images would improve the AI-based methods to predict the diagnosable material for histology (Figure 5). ResNet-34 [24], an image recognition model, was used as the image feature extraction device. One hundred and seventy-three specimens were classified into 145 specimens that were histologically diagnosable and 28 specimens that were histologically nondiagnosable, and we conducted an 8-fold cross-validation (13 h of training and 11 min of inference) on the 173 images. For training, we used the 3 networks surrounded by a square, and for inference, we used only the lower network surrounded by a dotted square, as shown in Figure 5.

### 2.9. Evaluation Items

The evaluation items were as follows: (1) comparison of AI-based methods using deep learning and MOSE by EUS experts for predicting diagnosable material for histology based on the stereomicroscopic images of EUS-FNB specimens, (2) comparison of diagnoses obtained by AI-based methods after contrastive learning and MOSE, (3) interobserver agreement among the endosonographers performing MOSE, (4) association between lesion features and MOSE positivity, and (5) evaluation of the tissue sample area and diagnostic accuracy.

### 2.10. Statistical Analysis

The sensitivity, specificity, accuracy, positive predictive value (PPV), and negative predictive value (NPV), were calculated to compare the performances of histological diagnoses predicted by the AI-based method and MOSE. Continuous variables are expressed as medians and interquartile ranges (IQRs). Categorical variables are expressed as percentages. The χ^2^ test was used to compare categorical parameters, and the Mann–Whitney U test was used to compare continuous variables. A *p*-value less than 0.05 was considered statistically significant. The cut-off value of the amount of specimen required for histological diagnosis was assessed by receiver operating characteristic (ROC) curve analysis, and the area under the ROC curve (AUC) was calculated. Interobserver agreement among the endosonographers performing MOSE was assessed using kappa statistics and defined as low (kappa coefficient, 0.01–0.20), fair (0.21–0.40), moderate (0.41–0.60), good (0.61–0.80), or excellent (0.81–1.00). All statistical analyses were performed using SPSS Statistics 27.0 (SPSS, Inc., Chicago, IL, USA).

## 3. Results

### 3.1. AI-Based Evaluation Using Deep Learning

The median age of the 63 patients was 65 years (IQR 58–72), 66.7% were male, and the median lesion size was 24 mm (IQR 20–35.5). The final diagnosis was PDAC in 41 cases, mass-forming pancreatitis (MFP) in 11 cases, autoimmune pancreatitis (AIP) in 8 cases, a pancreatic neuroendocrine tumour in 1 case, a metastatic pancreatic tumour in 1 case, and intraductal papillary mucinous carcinoma (IPMC) in 1 case (Table 1). The accuracy of the histological diagnoses per patient and per specimen with EUS-FNB was 82.5% (52/63) and 82.7% (81/98), respectively. The mean sensitivity, specificity, accuracy, PPV, and NPV of AlexNet with 3-fold cross-validation for obtaining a histological diagnosis were 85.8%, 55.2%, 71.8%, 69.5%, and 76.5%, respectively. In contrast, the corresponding values for MOSE were 88.9%, 47.1%, 81.6%, 88.9%, and 47.1%, respectively, showing that the diagnostic accuracy of the AI-based evaluation method using deep learning was not as high as that of MOSE performed by an EUS expert (Table 2). After a re-evaluation by increasing the number of specimens to 173, the accuracy of AI-based evaluation using deep learning was 74.5%, which showed a slight increase, however it was still not as good as that of MOSE by a human.

### 3.2. AI-Based Evaluation Using Contrastive Learning

The median age of the 96 patients was 68 years (IQR 60–74.75), 61.5% were male, and the median lesion size was 25 mm (IQR 20–35). The final diagnosis was PDAC in 66 cases, MFP in 13 cases, AIP in 11 cases, a pancreatic neuroendocrine tumour in 3 cases, a metastatic pancreatic tumour in 2 cases, and IPMC in 1 case (Table 1).

As shown in Figure 6, we succeeded in approaching the features of the stereomicroscopic images and the HE-stained images using contrastive learning in the feature space. The sensitivity, specificity, accuracy, PPV, and NPV of MOSE for obtaining a histological diagnosis were 88.9%, 53.5%, 83.2%, 90.8%, and 48.4%, respectively, whereas those of the AI-based diagnostic method using contrastive learning were 90.3%, 53.5%, 84.4%, 90.9%, and 51.7%, respectively (Table 3).

### 3.3. Interobserver Agreement among the Endosonographers Performing MOSE

According to the macroscopic evaluation performed by another expert endosonographer (HS) based on the stereomicroscopic images, the accuracy of this endosonographer for obtaining a histological diagnosis was 82.1% (141/173), which was similar to the 83.2% (144/173) demonstrated by the on-site endosonographer (TIs). Kappa coefficients showed that the interobserver agreement between the two readers was good (kappa coefficient = 0.612), but there were 19 cases (11%) with discrepant findings from the macroscopic evaluation.

### 3.4. Association between Lesion Features and MOSE Positivity

The median lesion size was significantly larger in cases judged as MOSE-positive (25 mm (IQR 22–34) vs. 21 mm (IQR 15–25), *p* < 0.001), and when the final diagnosis was divided into PDAC and non-PDAC, the proportion of PDAC was significantly higher in cases judged as MOSE-positive (69.7% (99/142) vs. 48.4% (15/31), *p* = 0.023).

### 3.5. Evaluation of Tissue Sample Area and Diagnostic Accuracy

The median tissue area of all 173 FNB specimens was 1.7 mm^2^ (IQR 0.8–2.92), and the tissue area was significantly larger in patients who were correctly diagnosed by histology (2.12 mm^2^ (IQR 1.11–3.18) vs. 0.465 mm^2^ (IQR 0.17–0.81), *p* < 0.001). In the ROC curve analysis, the AUC was 0.879, and the cut-off value of the tissue area for histological diagnosis calculated based on Youden’s index was 1.05 mm^2^, with a sensitivity of 78.6% and a specificity of 89.3%.

## 4. Discussion

In this study, we aimed to develop a new evaluation method for EUS-FNB specimens in pancreatic diseases, and AI-based evaluation using contrastive learning showed a diagnostic performance as good as that of MOSE performed by EUS experts.

EUS-FNA for the pancreas was first reported by Vilmann et al., in 1992 [1], and there have been many reports showing its usefulness and safety in obtaining tissue from the pancreas [4,25,26,27]. The major challenge of EUS-FNA/FNB is how accurately we can estimate whether an appropriate specimen is obtained at the time of the procedure since the specimen is basically obtained using a thin needle. MOSE is a recently introduced method to estimate the quality of the obtained specimen. Iwashita et al. [9] first assessed the efficacy of MOSE in estimating the adequacy of histologic core specimens obtained by EUS-FNA with a standard 19-gauge needle for solid lesions and concluded that a macroscopically visible core of ≥4 mm on MOSE could be an indicator of specimen adequacy and lead to an improved diagnostic yield. Since then, there have been several reports showing the usefulness of MOSE in EUS-FNB using core needles [28,29,30,31]. However, since there is no established method for MOSE and the judgement is subjective, the diagnostic ability of this approach may vary depending on the facility and the endosonographer. In the present study, the accuracy of MOSE performed independently by two EUS experts was 83.2% and 82.1%, showing a high performance for both endosonographers, and the interobserver agreement rate was good (kappa coefficient = 0.612). However, 19 of 173 specimens (11%) showed discrepant results. In these 19 specimens, they tended to include a lot of blood components, and even if there were whitish core specimens, they were difficult to distinguish due to small amounts or overlapping with blood clots. Thus, we aimed to develop a more objective and reproducible evaluation method for EUS-FNB specimens using AI.

This study is unique in that we used stereomicroscopic images of EUS-FNB specimens for MOSE and AI-based diagnosis. We believe that the use of a stereomicroscope has several advantages. First, compared with normal macroscopic observation, simply magnifying the image makes it easier to recognise blood clots and core tissue. In addition, all specimens can be evaluated under the same conditions by aligning the magnification degree using a scale under stereomicroscope observation. This increases the objectivity of the evaluation and leads to image-based analyses using AI.

We first examined an AI-based diagnostic method using deep learning, AlexNet, simply by using the images of fresh specimens obtained from EUS-FNB. AlexNet was the first architecture to incorporate the concepts of deep learning and CNNs for object recognition [21]. AlexNet achieved breakthrough performance in the Image Classification Challenge Contest (ILSVRC). In the image classification challenge contests before 2012, feature values were extracted from images and used to classify the images. At that time, when features were extracted from images, the designed features included colour, brightness, and object shape. Therefore, the performance of image classification depended on how effectively the features could be designed. However, AlexNet showed that the machine itself can find features without human input as long as enough data is available. Several concepts first introduced in AlexNet have influenced the development of subsequent image classification architectures and have become current standard techniques: AlexNet incorporated deep learning for image classification by connecting convolutional layers and multiple neural networks, and the ReLU function was adopted as the activation function of the neural network. In addition, to address the issue of a small amount of imaging data, augmentation was performed by flipping and rotating the existing image data. In the present study, the AI-based diagnostic method using AlexNet showed an acceptable accuracy of 71.8% for obtaining a histological diagnosis, however this method was not as good as MOSE performed by endosonographers. In addition, although the accuracy increased slightly by increasing the number of specimens, it was still not as good as that of MOSE by a human, and it was expected that deep learning using only stereomicroscopic images with such a small number of specimens and unbalanced sample size between diagnosable and undiagnosable cases, as used in this study, would not be able to achieve sufficient accuracy.

In reviewing the images used for AI analysis with deep learning, we found that images with whitish areas but a small amount of specimen tended to be incorrectly predicted with undiagnosable material for histology, and images with large amounts of blood components tended to be incorrectly predicted with diagnosable material for histology. Therefore, we attempted to reduce the attention given to the amount of specimen by refraining from the image reduction in the network and making the network learn to move away from images that can be correctly predicted and those that cannot be correctly predicted in the feature space. In addition, since it is relatively easy to estimate the amount of tissue in HE-stained specimens, we aimed to improve the diagnostic performance of the AI system by linking stereomicroscopic images and HE-stained images using contrastive learning methods, such as SimCLR [23]. SimCLR simplifies the self-supervised learning algorithms that have been proposed in recent years without the need for special architectures or memory banks. Most of the current AI creations are based on supervised learning, in which a human prepares a large amount of correct data and then allows the AI to start learning. Unsupervised learning is the concept of a system that requires drastically less training by humans and allows the AI to learn more on its own. As a result, we succeeded in bringing the features of the stereomicroscopic and HE-stained images close to each other and obtained a diagnostic performance equivalent to that of MOSE performed by EUS experts.

Previous studies have reported that there is a positive correlation between the tissue volume obtained by EUS-FNB and the performance of the histological diagnosis [9,28,32]. Similar to previous reports, there was a positive correlation between tissue volume and histological diagnostic performance with 1.05 mm^2^ as the cut-off value. Therefore, in the future, we are planning to use a multistep discrimination method, in which the amount of tissue is estimated first, and then the possible histological diagnoses are evaluated in a group with a small amount of tissue to surpass the accuracy of the macroscopic evaluation performed by humans.

This study has several limitations. First, the number of patients included in this study was small, and further studies with a larger number of patients are necessary to validate our results. Second, there might be discrepancies between the pathological diagnosis of EUS-FNB specimens and the actual pathological diagnosis of tumours that were not surgically resected. However, in these cases, immunohistochemical analysis results and/or periodic follow-up examinations may have minimised the discrepancies. Finally, all images, including those in the test datasets, were obtained retrospectively from a single hospital without randomization, making it difficult to exclude selection bias.

## 5. Conclusions

In conclusion, the AI-based method using contrastive learning to evaluate stereomicroscopic images of EUS-FNB specimens showed a diagnostic performance comparable to that of MOSE performed by EUS experts. However, it cannot be determined at this time whether our method can really be beneficial as an alternative to MOSE, mainly due to the insufficient number of subjects. Further studies with a better design and a larger sample size are necessary to validate our results. Incorporating information on the amount of tissue into the current AI system may help further improve the diagnostic performance of this approach and establish a new evaluation method for EUS-FNB specimens that can be an alternative to MOSE.

## Figures and Tables

**Figure 1 diagnostics-12-00434-f001:**
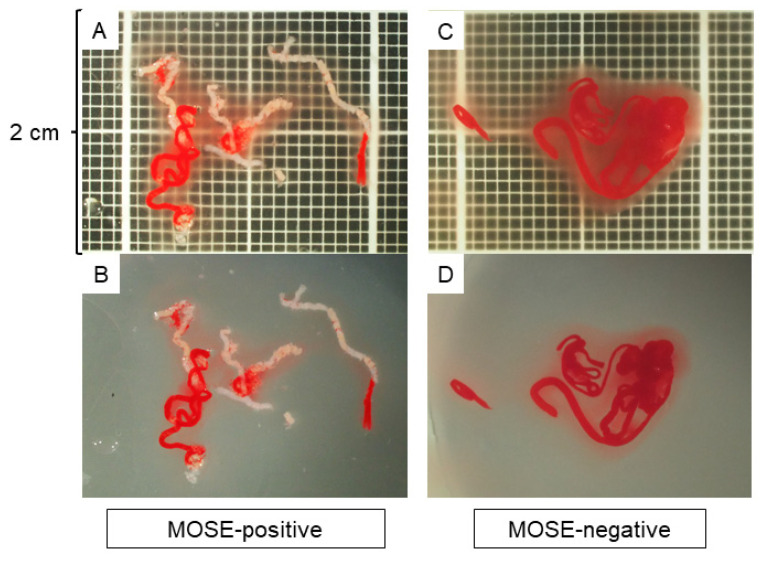
Macroscopic on-site evaluation (MOSE) using a stereomicroscope. The observation screen was set up with a vertical width of 2 cm with a scale showing 1-mm increments in the background. The quality of the specimen was evaluated without anything in the background, and the positivity of the MOSE results was judged based on the presence of whitish core tissue. (**A**,**B**): A specimen judged as MOSE-positive with (**A**) and without (**B**) a scale in the background. (**C**,**D**): A specimen judged as MOSE-negative with (**C**) and without (**D**) a scale in the background.

**Figure 2 diagnostics-12-00434-f002:**
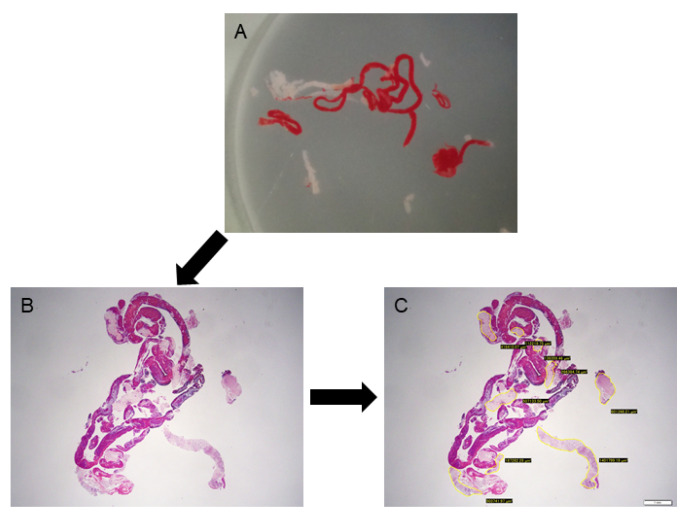
Evaluation of the tissue area using imaging software. (**A**): A stereomicroscopic image of an endoscopic ultrasound-guided fine-needle biopsy specimen. (**B**): Haematoxylin and eosin staining of the specimen, viewed in a low-power field. (**C**): Measuring the area of the tissue specimen, excluding blood clots, using imaging software (CellSense).

**Figure 3 diagnostics-12-00434-f003:**
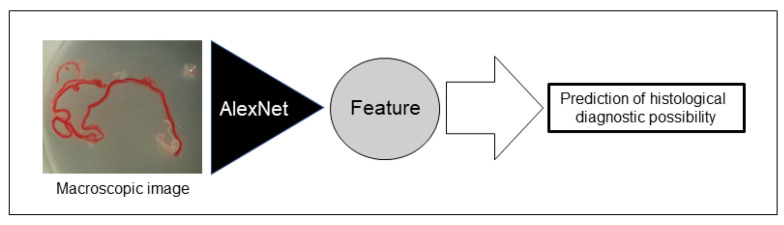
Schema of the deep learning network used for this study. Artificial intelligence (AI) extracted imaging features from stereomicroscopic images, and we examined whether AI could predict the possible histological diagnosis based on stereomicroscopic images.

**Figure 4 diagnostics-12-00434-f004:**
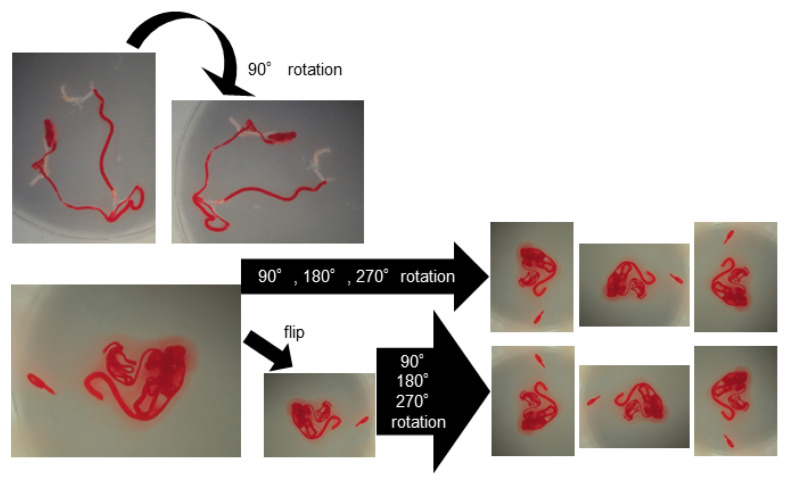
Data augmentation of the images for deep learning. The histologically diagnosable images were augmented through a 90-degree rotation, and the nondiagnosable images were augmented by flipping plus 3-way rotation.

**Figure 5 diagnostics-12-00434-f005:**
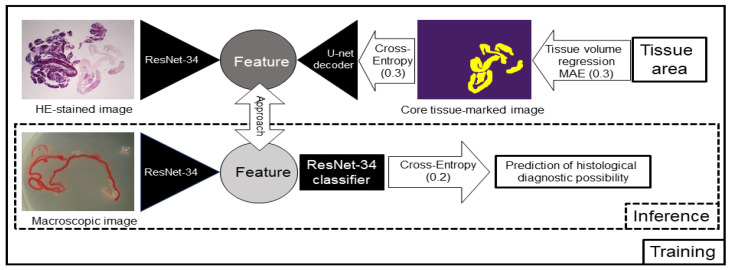
Schema of the contrastive learning network used for this study. Artificial intelligence (AI) learned the relationships between the haematoxylin and eosin (HE)-stained image and the image with the marked core tissue and detected which pixels corresponded to the tissue. The contrastive learning method was also trained to approach the features of the stereomicroscopic images and HE-stained images, and we investigated whether linking the two images would improve the prediction rate of the AI-based diagnostic method for a positive histological diagnosis. For training, we used the 3 networks surrounded by a square, and for inference, we used only the lower network surrounded by a dotted square. MAE: mean absolute error. The numbers in parentheses represent the weight of each loss.

**Figure 6 diagnostics-12-00434-f006:**
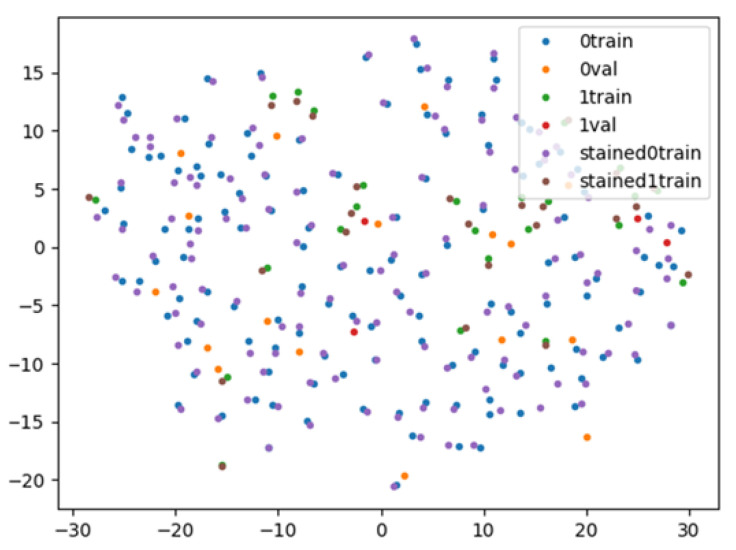
Visualisation of the feature space by contrastive learning. The stereomicroscopic/haematoxylin and eosin (HE)-stained images with positive histological diagnoses indicated by blue/purple points are close to each other in the feature space. Blue points: diagnosable stereomicroscopic images for training; orange points: diagnosable stereomicroscopic images for validation; green points: undiagnosable stereomicroscopic images for training; red points: undiagnosable stereomicroscopic images for validation; purple points: diagnosable HE-stained images; and brown points: undiagnosable HE-stained images.

**Table 1 diagnostics-12-00434-t001:** Patient Characteristics.

	Deep Learning Group	Contrastive Learning Group
	N = 63	N = 96
Age, median (IQR)	65 (58–72)	68 (60–74.75)
Sex, Male, N (%)	42 (66.7)	59 (61.5)
Lesion size, median (IQR), mm	24 (20–35.5)	25 (20–35)
Final diagnosis, N (%)		
Pancreatic ductal adenocarcinoma	41 (65.1)	66 (68.8)
Mass-forming pancreatitis	11 (17.5)	13 (13.5)
Autoimmune pancreatitis	8 (12.7)	11 (11.5)
Pancreatic neuroendocrine tumour	1 (1.6)	3 (3.1)
Pancreatic metastasis	1 (1.6)	2 (2.1)
Intraductal papillary mucinous carcinoma	1 (1.6)	1 (1.0)

IQR: interquartile range.

**Table 2 diagnostics-12-00434-t002:** Diagnostic performances of the AI-based evaluation method using deep learning and MOSE for obtaining a histological diagnosis.

		Histology					Histology		
		Diagnosable	Undiagnosable	Total			Diagnosable	Undiagnosable	Total
AI	Diagnosable	139	23	162	MOSE	Diagnosable	72	9	81
	Undiagnosable	61	75	136		Undiagnosable	9	8	17
	Total	200	98	298		Total	81	17	98
	Sensitivity	85.8%				Sensitivity	88.9%		
	Specificity	55.2%				Specificity	47.1%		
	Accuracy	71.8%				Accuracy	81.6%		
	PPV	69.5%				PPV	88.9%		
	NPV	76.5%				NPV	47.1%		

AI: artificial intelligence, MOSE: macroscopic on-site evaluation, PPV: positive predictive value, NPV: negative predictive value.

**Table 3 diagnostics-12-00434-t003:** Diagnostic performances of the AI-based evaluation method using contrastive learning and MOSE for obtaining a histological diagnosis.

		Histology					Histology		
		Diagnosable	Undiagnosable	Total			Diagnosable	Undiagnosable	Total
AI	Diagnosable	131	13	144	MOSE	Diagnosable	129	13	142
	Undiagnosable	14	15	29		Undiagnosable	16	15	31
	Total	145	28	173		Total	145	28	173
	Sensitivity	90.3%				Sensitivity	88.9%		
	Specificity	53.5%				Specificity	53.5%		
	Accuracy	84.4%				Accuracy	83.2%		
	PPV	90.9%				PPV	90.8%		
	NPV	51.7%				NPV	48.4%		

AI: artificial intelligence, MOSE: macroscopic on-site evaluation, PPV: positive predictive value, NPV: negative predictive value.

## Data Availability

The data used to support the findings of this study are available from the corresponding author upon request.

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
