# Peer review of "Development of a Novel Evaluation Method for Endoscopic Ultrasound-Guided Fine-Needle Biopsy in Pancreatic Diseases Using Artificial Intelligence"

_diagnostics, 2022, doi:10.3390/diagnostics12020434_

Round 1

Reviewer 1 Report

The authors attempted to develop a novel evaluation method for endoscopic ultrasound-guided fine-needle biopsy in pancreatic disease using artificial intelligence (AI). They concluded that the AI-based method using contrastive learning to evaluate stereomicroscopic images of specimens showed a diagnostic performance comparable to that of macroscopic on-site evaluation. Their results may lead to future AI prospects in endoscopic ultrasound-guided fine-needle biopsy. However, it cannot be determined at this time whether their method can really be beneficial as an alternative to macroscopic on-site evaluation as the authors said in their conclusion, mainly due to the insufficient number of subjects. I look forward to a study with a better design and a larger sample size in the future.

Author Response

Response to Reviewer 1 Comments

Point 1: The authors attempted to develop a novel evaluation method for endoscopic ultrasound-guided fine-needle biopsy in pancreatic disease using artificial intelligence (AI). They concluded that the AI-based method using contrastive learning to evaluate stereomicroscopic images of specimens showed a diagnostic performance comparable to that of macroscopic on-site evaluation. Their results may lead to future AI prospects in endoscopic ultrasound-guided fine-needle biopsy. However, it cannot be determined at this time whether their method can really be beneficial as an alternative to macroscopic on-site evaluation as the authors said in their conclusion, mainly due to the insufficient number of subjects. I look forward to a study with a better design and a larger sample size in the future.

Response 1: Thank you very much for your informative comments. We agree that the small number of subjects is a limitation of our study. We added comments in the conclusion bases on your comments as follows; “However, it cannot be determined at this time whether our method can really be beneficial as an alternative to MOSE, mainly due to the insufficient number of subjects. Further studies with a better design and a larger sample size are necessary to validate our results.” (Page 15, line 471-Page 16, line 474)

Reviewer 2 Report

This is a retrospective study aimed at evaluating a new artificial intelligence method to evaluate the histological adequacy using artificial intelligence.

Overall, the manuscript is well-written, and the methodology is correct. A very interesting study.

Introduction: you included a paragraph about the possibility to avoid ROSE during EUS-FNB. For clarity for readers, I think that a study reporting the technique to perform ROSE on EUS-FNB specimens should be mentioned (PMID: 30484917).

Figure 1: please add letters and explanations at the four panels of Figure 1

Methods: Please specify how the final diagnosis was assessed.

Results: did you find any association between lesion features (e.g., size or diagnosis, PDAC vs non-PDAC) and MOSE adequacy?

Author Response

Response to Reviewer 2 Comments

This is a retrospective study aimed at evaluating a new artificial intelligence method to evaluate the histological adequacy using artificial intelligence.

Overall, the manuscript is well-written, and the methodology is correct. A very interesting study.

Point 1. Introduction: you included a paragraph about the possibility to avoid ROSE during EUS-FNB. For clarity for readers, I think that a study reporting the technique to perform ROSE on EUS-FNB specimens should be mentioned (PMID: 30484917).

Response 1. Thank you very much for your comment. We added comments regarding the touch imprint cytology technique for EUS-FNB specimens in the Introduction section as follows and cited the article as a reference; “Recently, a touch imprint cytology technique was reported for EUS-FNB specimens, which allows to obtain both cytological and histological specimens at the same time with the same needle, as well as to perform ROSE. This technique provided comparable samples to those of EUS-FNA-standard cytology, and combined the benefits of cytology and histology for the evaluation.” (Page 2, lines 52-56).

Point 2. Figure 1: please add letters and explanations at the four panels of Figure 1

Response 2. Thank you for raising this point. We added letters and explanations at the four panels of Figure 1 (Page 4, lines 169-171).

Methods: Please specify how the final diagnosis was assessed.

Response 3. Thank you for your comment. In our study, the final diagnosis was based on the surgical specimen or the clinical course consistent with the results of EUS-FNB with a minimum 6-month follow-up. We added the sentence to describe how the final diagnosis was assessed in the Methods section (Page 5, 178-180).

Results: did you find any association between lesion features (e.g., size or diagnosis, PDAC vs non-PDAC) and MOSE adequacy?

Response 4. Thank you very much for your comment. As you suggested, the median lesion size was significantly larger in cases judged as MOSE-positive (25 mm [IQR 22-34] vs. 21 mm [IQR 15-25], P<0.001), and when the final diagnosis was divided into PDAC and non-PDAC, the proportion of PDAC was significantly higher in cases judged as MOSE-positive (69.7% [99/142] vs. 48.4% [15/31], P=0.023). We added these results as a result 3.4. (Page 13, lines 362-366).
